

# Noise performance of microwave humidity sounders over their life time

Imke Hans[1], Martin Burgdorf[1], Viju O. John[2,3], Jonathan Mittaz[4,5], and Stefan A. Buehler[1]

[1]Meteorologisches Institut, Centrum für Erdsystem- und Nachhaltigkeitsforschung (CEN), Universität Hamburg, Bundesstrasse 55, 20146 Hamburg, Germany
[2]European Organisation for the Exploitation of Meteorological Satellites, Eumetsat Allee 1, D-64295 Darmstadt, Germany
[3]Met Office Hadley Centre, FitzRoy Road, Exeter, Devon EX1 3PB, United Kingdom
[4]Department of Meteorology, University of Reading, Reading, RG6 6AL, UK
[5]National Physical Laboratory, Teddington, TW11 0LW, UK

*Correspondence to:* Imke Hans (imke.hans@uni-hamburg.de)

**Abstract.** The microwave humidity sounders Special Sensor Microwave Water Vapour Profiler (SSMT-2), Advanced Microwave Sounding Unit-B (AMSU-B) and Microwave Humidity Sounder (MHS) to date have been providing data records for 25 years. So far, the data records lack uncertainty information essential for constructing consistent long time data series. In this study, we assess the quality of the recorded data with respect to the uncertainty caused by noise. We calculate the noise on the raw calibration counts from the deep space view (DSV) of the instrument and the Noise Equivalent Differential Temperature (NE$\Delta$T) as a measure for the radiometer sensitivity. For this purpose, we use the Allan Deviation that is not biased from an underlying varying mean of the data and that has been suggested only recently for application in atmospheric remote sensing. Moreover, we use the bias function related to the Allan Deviation to infer the underlying spectrum of the noise. As examples, we investigate the noise spectrum in flight for some instruments. For the assessment of the noise evolution in time, we provide a descriptive and graphical overview of the calculated NE$\Delta$T over the life span of each instrument and channel. This overview can serve as an easily accessible information for users interested in the noise performance of a specific instrument, channel and time. Within the time evolution of the noise, we identify periods of instrumental degradation, which manifest themselves in an increasing NE$\Delta$T, and periods of erratic behaviour, which show sudden increases of NE$\Delta$T interrupting the overall smooth evolution of the noise. From this assessment and subsequent exclusion of the aforementioned periods, we present a chart showing available data records with NE$\Delta$T < 1 K. Due to overlapping life spans of the instruments, these reduced data records still cover without gaps the time since 1994 and may therefore serve as first step for constructing long time series. Our method for count noise estimation, that has been used in this study, will be used in the data processing to provide input values for the uncertainty propagation in the generation of a new set of Fundamental Climate Data Records (FCDR) that are currently produced in the project "Fidelity and Uncertainty in Climate data records from Earth Observation (FIDUCEO)".



# 1  Introduction

In this study, we calculate and assess the noise evolution over the life time of all individual instruments of the microwave sounders Special Sensor Microwave Water Vapour Profiler (SSMT-2), Advanced Microwave Sounding Unit-B (AMSU-B) and Microwave Humidity Sounder (MHS). So far, their data sets lack comprehensive information on uncertainty caused by noise:

From the pre-launch measurements, one knows the specifications on the accuracy that the instruments had to meet. These values of Noise Equivalent Differential Temperature (NE$\Delta$T) are provided per instrument and channel in the NOAA KLM User Guide (Robel et al., 2009) and by their nature as specifications do not comprise any information on time evolution of noise. The ATOVS and AVHRR Pre-processing Package (AAPP) software used for the processing of raw level 1b data to level 1c data containing brightness temperatures, now provides with version 7.13 a measure of noise, namely a cold and a warm

NE$\Delta$T, referring to the cold and warm calibration targets on board those microwave sounders. However, this information on noise in the AAPP-processed data sets is not available for all instruments. Graphical information on noise evolution is given on the NOAA-STAR-ICVS web-page (NOAA, 2015), but this is also limited to a few periods and instruments. Comprehensive information on uncertainty caused by noise is not available for the end user interested in the measurements of the SSMT-2, AMSU-B or MHS instruments.

To close this gap, we determine and evaluate the time series of the noise for the SSMT-2 instruments on board the Defense Meteeorological Satellite Program (DMSP) satellites F11, F12, F14 and F15, for the AMSU-B instruments on the satellites NOAA-15, -16 and -17 launched by the National Oceanic and Atmospheric Administration (NOAA) and for the MHS instruments on the satellites NOAA-18,-19 and the Metop-A and -B satellites controlled by EUMETSAT. In the assessment of the noise evolution, we identify periods of low quality data. To make this information easily accessible, we provide a graphical and

descriptive overview over the whole life time of the instruments. From this overview, the user can estimate the uncertainty due to noise and he or she can decide on the applicability of the data set for his purposes. Our method and tool to estimate count noise will be used in the evaluation of the uncertainty for the generation of new microwave sounder Fundamental Climate Data Records (FCDR). Those are currently developed in the project "Fidelity and Uncertainty in Climate data records from Earth Observation (FIDUCEO)" in the framework of which this study has been carried out and that aims to adopt a rigorous

metrological (measurement science) perspective to understanding the origins and quantifying various instrumental issues that lead to random and systematic errors (Mittaz et al., 2017).

Apart from the new comprehensive time series of noise evolution, our results also include the analysis of the spectrum of the noise in flight. This analysis is based on the statistical tool of the Allan Deviation and its general form the M-sample-Deviation (Mittaz, 2016). We also use the Allan Deviation for the calculation of the noise itself, in contrast to what has been done for

the previously available noise estimates. The Allan Deviation, well known in other disciplines (Tian et al., 2015; Malkin, 2011; Allan, 1966), has been suggested only recently by Tian et al. (2015) for the estimation of noise in the measurements of microwave sounders in flight.

The noise in flight can be estimated with various methods. Atkinson (2015) reports on methods used and suggested by different agencies for the calculation of cold and warm NE$\Delta$T. The various methods include the standard deviation and also





the Allan Deviation as suggested by Tian et al. (2015). The disadvantage of the standard deviation is that it is sensitive to variations in the mean that naturally occur in the measurements of these kind of polar orbiting instruments (Tian et al., 2015). In this study, we follow the suggestion of Tian et al. and use the Allan Deviation for the estimation of noise. To clarify the notion of noise at first, the next section is dedicated to the elaboration of a consistent noise terminology in the context of the

microwave sounders.

This article is further structured as follows: After establishing the noise terminology used here, we explain our methods and data in detail. Later, our results on the analysis of the noise spectra and the time evolution of noise are presented. The discussion of these findings is followed by concluding thoughts. In the appendix we provide a detailed description of the time series of the individual instruments.

## 2   Noise terminology

In theory, noise in the measurements of a radiometer such as the microwave sounders considered here can be related to the process of measuring and it can be calculated from instrumental quantities. This theory of noise in the measurements of a radiometer is explained in Ulaby and Long (2014) whom we follow here: The antenna delivers a power $P_a$ to the receiver. In the Rayleigh-Jeans limit, this power is usually related to a temperature $T_a$ as $P_a = kT_aB$, with $k$ being Boltzmann's constant

and $B$ the bandwidth of the receiver. The precision with which the temperature $T_a$ can be estimated by a measurement is referred to as the radiometer sensitivity $dT$. It is subject to any noise that may impact on the true signal and depends on the temperature of the whole system. So, the total system noise power $P_{sys} = P_a + P_{rec} = kT_{sys}B$ relates to the system temperature $T_{sys} = T_a + T_{rec}$ with $T_a$ being the antenna temperature (which includes the true signal) and $P_{rec}$ and $T_{rec}$, being the power and temperature of the receiver including the influence of the transmission line between antenna and receiver. Since

the final measured output voltage is an integrated value from a receiver of bandwidth $B$ and an integration time of $t$, the noise uncertainty to the radiometer sensitivity is

$$dT_N = \frac{T_{sys}}{\sqrt{Bt}} \tag{1}$$

However, one also has to consider fluctuations in the gain $G$ on time-scales shorter than one calibration cycle. These are not calibrated out, but impact on the recorded voltage and hence lead to fluctuations in the final measurement result. These short

term gain fluctuations lead to a term

$$dT_G = T_{sys}\frac{dG}{G} \tag{2}$$

Since both contributions are independent, the radiometer sensitivity finally reads

$$dT = \sqrt{dT_N^2 + dT_G^2} \tag{3}$$

$$\Rightarrow dT = T_{sys} \cdot \sqrt{\frac{1}{Bt} + \left(\frac{dG}{G}\right)^2} \tag{4}$$



$T_{sys}$ is the sum of antenna temperature and combined receiver-transmission line temperature. This radiometer sensitivity $dT$ describes the smallest temperature difference that the radiometer can distinguish when looking at a target inducing an antenna temperature of $T_a$. It is therefore an uncertainty estimate on the measurement of $T_a$.

For in-flight monitoring of the radiometer sensitivity, eq. 4 is not well suited, since the receiver-transmission line temperature

is not well accessible. Therefore, one does not usually use the above eq. 4 to calculate the radiometer sensitivity, but one uses some kind of statistical estimation of the fluctuations in the measurements, e.g. in the counts that are the digitized output voltage. The counts may stem from the instrument's view of the cold or warm calibration target (deep space view, DSV, and on-board calibration target, OBCT). This statistical estimation may be the standard deviation or the Allan Deviation. This estimation of the fluctuations in the counts, referred to as count-noise, comprises every noise that has contaminated

the true signal from the antenna over the transmission line through amplifiers and mixers, including digitization noise. The count-noise is therefore subject to both effects described in eq. 4 for the total radiometer sensitivity: noise from all electronic devices, that remains due to a finite integration time, and short-term gain fluctuations (on time scales shorter than one scan, i.e. one calibration cycle). This count-noise estimate is in units of counts, i.e. one cannot compare the values directly among different sensors since the absolute count values are somewhat arbitrary. However, we can transform the count-noise into a noise

equivalent differential temperature (NEΔT), which then represents an estimate for the total radiometer sensitivity described by eq. 4. This transformation includes the gain, i.e. NEΔT = noise-in-counts/ gain. The actual value of the gain taken for this estimate, is the one corresponding to the scan lines from which the count-noise has been calculated.

Hence, this transformation translates the fluctuations that we see in the counts (count-noise) into a temperature difference that is equivalent to the noise by using the current gain. Any long-term changes in the gain will therefore impact on the time

evolution of NEΔT. Altogether, this NEΔT includes the actual noise, i.e. short term fluctuations of whatever origin, and the long-term variations of the gain. This is in contrast to the count-noise estimate which reflects the pure short-term fluctuations.

This in-orbit analysis of noise can be carried out on both the DSV and OBCT view counts, whereas the earth counts are not suited due to their natural variability when scanning over different scenes on earth. The choice of target influences the antenna temperature in eq. 4 and consequently influences $dT$. Therefore, the $dT$ calculated from the OBCT is expected to be larger

than from the DSV. If choosing the DSV counts, one takes advantage of the fact that the brightness temperature of the DSV is very low. Therefore, the contribution of that signal to the antenna temperature is rather weak. The remaining contribution to the antenna temperature and of course the receiver temperature are of instrumental origin. Hence, analysing the DSV will give results (almost only) on the instrument itself. Converting the DSV-count-noise to a temperature we obtain the cold NEΔT, which corresponds to the radiometer sensitivity when looking at very cold scenes. The second choice, taking the count-noise

on OBCT will lead to the warm NEΔT after translating to a temperature. This warm NEΔT corresponds to the radiometer sensitivity when looking at a target of approximately 280 to 300 K (temperature of the OBCT, $T_{\mathrm{BB}}$).

The end-user, however, will be interested in the (scene) NEΔT that he or she has to expect for a certain earth pixel in his or her dataset. However, the NEΔT cannot be calculated directly from the earth counts as explained above. But, it can be estimated from the cold and warm NEΔT. As eq. 4 expresses, the NEΔT or radiometer sensitivity depends on the antenna

temperature. Therefore, the NEΔT when looking at an earth scene of 240 K will be close to but slightly smaller than the warm





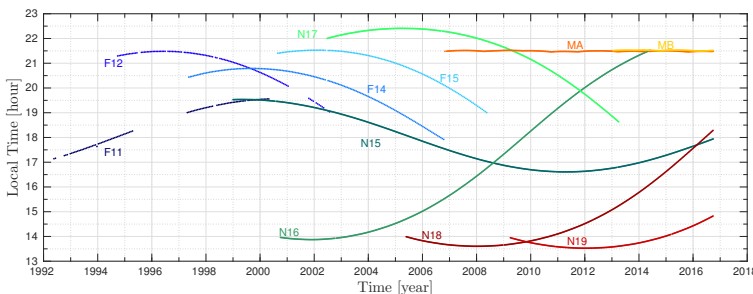

**Figure 1.** The local equator crossing times (ascending branch) for the satellites considered here. The graphic shows all times for which there is any data available from the instrument on board the respective satellite (regardless of quality issue). For the F11 to F15 satellites, we only look at part of the time series shown here, due to inconsistencies across the different sources of data (see text).

NE$\Delta$T. Knowing both the cold and the warm NE$\Delta$T, the user can calculate his or her scene NE$\Delta$T as

$$\text{NE}\Delta\text{T} = \text{NE}\Delta\text{T}_{cold} + (T_{\text{a,scene}} - T_{\text{a,DSV}}) \cdot m \tag{5}$$

$$\text{with } m = \frac{\text{NE}\Delta\text{T}_{warm} - \text{NE}\Delta\text{T}_{cold}}{T_{\text{a,OBCT}} - T_{\text{a,DSV}}}$$

$$\text{and } T_{\text{a,OBCT}} - T_{\text{a,DSV}} = T_{\text{BB}} - 2.725K$$

5    This equation is obtained from combining eq. 4 applied for warm and cold NE$\Delta$T (i.e. using $T_{\text{a,OBCT}}$ and $T_{\text{a,DSV}}$ as antenna temperatures in $T_{\text{sys}}$). Using the resulting two equations for eq. 4 with scene NE$\Delta$T will yield the above equation 5.

To obtain an estimate of the current scene NE$\Delta$T, the user needs the estimate for the cold and warm NE$\Delta$T corresponding to the time window where his or her current earth pixel belongs. Moreover, he or she needs the corresponding temperature of the OBCT (measured temperature of the black body, $T_{\text{BB}}$) and the $T_{a,scene}$ of his or her data set to finally calculate scene-NE$\Delta$T

10   with eq. 5.

## 3   Data and Methods

### 3.1   Microwave sounder data

This study covers a time range of 22 years from 1994 to 2016 of microwave data. We investigate the life time stability of the polar orbiting satellites DMSP-F11, 12, 14, 15, the NOAA-15 to NOAA-19 and the Metop-A and -B satellites. All of them are

15   placed in sun-synchronous orbits, but over the life time of most of the satellites the orbits drift nonetheless and therefore have a changing local equator crossing time (LECT) (Ignatov et al., 2004). Only the Metop satellites have a constant local ECT due to their controlled orbit. An overview of the ECT over the missions of the satellites is shown in Fig. 1. The drifting as well as the constant orbits of the Metop satellites are clearly visible.





The microwave sensors on board those satellites are the Special Sensor Microwave Water Vapor Profiler (SSMT-2), the Advanced Microwave Sounder Unit - B (AMSU-B) and the Microwave Humidity Sounder (MHS) instruments. Table 1 in Kobayashi et al. (2016) shows the characteristics of those sounders. The instruments are cross-track scanners with a very similar calibration cycle: after four views on the OBCT, 90 earth views are scanned before four views on the deep space are

recorded. This procedure is continuously repeated with a duration of 8/3 seconds. Therefore, every 8/3 seconds, i.e. every scan, a new calibration of the instrument with OBCT and DSV is carried out. In the following, we describe the location of the channels for the different instruments. Note that the five AMSU-B channels are counted including the AMSU-A channels, therefore they are correctly named as channels 16-20. However, since their order is the same as for MHS with channels 1-5, we keep the 1-5 naming for simplicity. For AMSU-B and MHS, the water vapour sensitive channels are channel 3-5 with

frequencies 183±1, 183±3 and 183±7 GHz (only 190 GHz for MHS) around the 183 GHz water vapour absorption line. They provide information on the tropospheric humidity. Channel 1 is at about 89 GHz and channel 2 at 150 GHz (157 GHz for MHS); both offer a deeper view through the atmosphere down to the surface. For SSMT-2, the order of the channels is different. The original channel 1 is at 183±3 GHz, channel 2 at 183±1 and channel 3 at 183±7 GHz. The original channels 4 and 5 are surface channels placed at 92 and 150 GHz, respectively. However, again for simplicity, we will use the MHS-naming

of channels for SSMT-2 as well and refer to the water vapour channels at 183±1, 183±3 and 183±7 GHz as channel 3,4 and 5. The surface channels at 92 and 150 GHz are labelled as channel 1 and 2. Note that the actual frequencies are not exactly the same for the different instruments, even though we will refer to them as one channel, e.g. the "89 GHz-channels" encompass the 89 GHz channels of AMSU-B and MHS, but also the 92 GHz channel of SSMT-2.

We use binary level-1b data sets downloaded from the NOAA CLASS (Comprehensive Large Array-data Stewardship Sys-

tem) archive. For SSMT-2 there are some inconsistencies regarding the time range of available data: On the NOAA NCEI (National Centers for Environmental Information, formerly NGDC, National Geophysical Data Center) data availability webpage, there are longer time frames indicated for which SSMT-2 data should exist (reaching back to 1992) than on the NOAA CLASS page. This larger data set of SSMT-2 data has been reformatted to NetCdf by John et.al. in (John and Chung, 2014) and covers the range according to NCEI (shown in Fig. 1). But this is not the raw file providing all information that goes into

the calibration and that we aim to look at. For example, the NCEI-data file does not contain the temperature measured on the internal black body. Hence, to stay in line with the investigation of AMSU-B and MHS data obtained from NOAA CLASS, we only used the binary data for SSMT-2 that NOAA CLASS provides and that cover the time range indicated on the NOAA CLASS website (NOAA-CLASS, 2016). The data record format for the binary level 1b data that we use here is documented for AMSU-B and MHS in the NOAA KLM User Guide (Robel et al., 2009) and on the NOAA CLASS website for SSMT-2

(NOAA-CLASS, 2016). From this raw data record we read the counts for the black body views, i.e. the on board calibration target views (OBCT), the counts for the deep space views and the counts for the temperature sensors (Platinum Resistance Thermometers, PRTs) on the black body. The latter ones are transformed to temperature in Kelvin. We do not take into account any quality flags that might be set in the data record but only use raw unfiltered data in order to preserve the original recorded behaviour. For each channel and all scan lines of every orbit we calculate from those values the gain $G_n(i)$ for scan line $n$ and





channel $i$ as

$$G_n(i) = \frac{\overline{C}_{\mathrm{OBCT}}(i) - \overline{C}_{\mathrm{DSV}}(i)}{\overline{T}_{\mathrm{PRT}} - 2.725\mathrm{K}}, \tag{6}$$

where $\overline{C}_{\mathrm{OBCT}}$ and $\overline{C}_{\mathrm{DSV}}$ indicate the counts from the OBCT and deep space view, respectively, both averaged over the 4 views. $\overline{T}_{\mathrm{PRT}}$ denotes the average temperature measured by all temperature sensors (2 for SSMT-2, 7 for AMSU-B and 5 for MHS).

The OBCT and DSV counts, as well as the gain, are our input values for the noise estimation which is described in the next section.

### 3.2   Noise estimation

The standard deviation can be used to estimate the noise on the counts as explained in the above section on noise terminolgy. This estimation has been used before (Tian et al., 2015; Atkinson, 2015), but the standard deviation has a disadvantage in the

context of noise monitoring of instruments on polar orbiting satellites: Since the standard deviation is based on measuring the difference of the values from the sample's mean, the standard deviation will only provide a sensible representation of the precision of the sample if the sample truly has a constant mean. However, due to the orbiting movement of the instrument around the earth, all measured quantities show orbital variations, i.e. they have a non-stationary mean over one orbit. For such cases, the standard deviation is biased as it expects a stationary mean and would measure the deviation of a single measured

value from the overall (erroneously stationary) mean over the full orbit. To reduce this bias, one has to define sub-samples along the orbit, for which the real mean is approximately stationary. Hence, the standard deviation becomes highly dependent on these chosen sample sizes and is therefore less suited for consistent in-orbit monitoring of different instruments.

    The Allan Deviation does not show this bias and is less dependent on choices (Tian et al., 2015; Mittaz, 2016). Therefore, we use the Allan Deviation as statistical tool to estimate the noise on the counts. The Allan Deviation, or its square, the Allan

Variance, is a special case of the more general M-sample Variance (Allan, 1966), that is defined as

$$\sigma_M^2(M) = \frac{1}{M-1}\left(\sum_{i=0}^{M-1} y_i^2 - \frac{1}{M}\left[\sum_{i=0}^{M-1} y_i\right]^2\right), \tag{7}$$

where $y_i$ is a measured value from the sample and $M$ denotes the number of values of the sample that are used for the calculation. That means, $M$ adjacent measurements yield one value $\sigma_{M_j}^2$. The associated total M-sample variance for a total sample of $N$ measurements is then calculated as the average over all $\sigma_{M_j}^2$ with $j \in 1, 2, ...N/(M-1) - 1$. With $M = 2$ in eq.

7, one obtains the Allan variance that effectively uses two adjacent measurements of the data series:

$$\sigma_{\mathrm{Allan}}^2 = \sigma_M^2(2) = \frac{1}{2}(y_1 - y_0)^2 \tag{8}$$

The total Allan Deviation for $N$ measurements is then written as

$$\sigma_{\mathrm{Allan, tot}} = \sqrt{<\sigma_{\mathrm{Allan}}^2>_N}$$

$$= \sqrt{\frac{1}{2(N-1)}\sum_{n=1}^{N-1}(y_{n+1} - y_n)^2} \tag{9}$$





It is not biased due to longer term trends such as orbital variations, because the Allan Deviation averages over a sample size $N$ the deviation of directly adjacent measurements. For a sample size of $N$ values, one computes $N-1$ Allan Deviations and averages these. The question of an appropriate value of $N$ has been investigated in (Tian et al., 2015) for the new instrument Advanced Technology Microwave Sounder (ATMS). Since it has the same scanning routine as the SSMT-2, AMSU-B and

MHS, we follow the suggestions of Tian et al.: The lower limit of $N$ is set by the stability of the Allan Deviation with changing $N$. For small sample sizes of $N<300$, the Allan Deviations fluctuates. From $N=300$ on, it takes a stable value. Following this study, we therefore use a sample size of $N=300$ scan lines, providing us with about 8 total Allan Deviations per orbit. As expected, by comparing the standard deviation with the Allan Deviation we found up to 40 times larger variations in the noise estimate over one orbit for the standard deviation. Also, increasing the sample size for the Allan Deviation does not

significantly change our results. This agrees with the results in (Tian et al., 2015) concerning the stabilization of the Allan Deviation above $N=300$ scan lines for those instruments.

For defining the number of $N$, one could also use a different approach. This relates to the question of what a single measurement is and what adjacent means in the context of the investigated instruments. As explained above, the instruments have a scanning and calibration cycle of 8/3 seconds during which they record the signal from four warm calibration target views,

from 90 earth views and from four deep space views. Between the different targets they record nothing. Having in mind this scanning and calibration cycle, there are two approaches of noise estimation using the Allan Deviation: one could use the Allan Deviation between the individual adjacent four calibration views. I.e. in each cycle one gets three Allan Deviations. Over one orbit one averages these $3 \cdot k$ Allan Deviations (with $k=$ number of scan lines in the orbit) to get the final total Allan Deviation. We will call this the inter-pixel-method. Opposed to that, one can act on the scale of scan lines, as usually done for noise inves-

tigations so far (Tian et al., 2015; Atkinson, 2015; EUMETSAT, 2013). For this, one calculates the Allan Deviation between two adjacent scan lines for all four views separately and then averages over the four obtained Allan Deviations before applying the average over $N$ scan lines. We chose this inter-scan-line-method with $N=300$ in our study. The reason is, that the results of our analysis of the noise spectrum speak in favour of this inter-scan-line-method for noise estimation (see section 4.1): The inter-scan-line-method will give a better estimate of the uncertainty in the data due to noise, compared to the inter-pixel-method

that underestimates the uncertainty for non-white noise spectra.

To analyse the noise spectrum, we make use of the Allan Deviation and the general M-sample variance again. Together, they build an interesting tool to determine the noise spectrum in a simple way (Allan, 1966, 1987; Barnes and Allan, 1990). The quotient of the M-sample variance and the Allan Variance, each averaged over the same sample size, is the so called bias function (Barnes and Allan, 1990)

$$B_1(M) = \frac{<\sigma_M^2>_N}{<\sigma_{\text{Allan}}^2>_N}. \qquad (10)$$

The behaviour of $B_1(M)$ for varying $M$ is characteristic for different noise spectra. We let $M$ vary from 2 to 20. We simulate white and pink noise in MATLAB and determine their bias functions over the indicated range of $M$. This serves as comparison tool for the bias functions obtained from real data to estimate the nature of the noise spectrum of the data. This spectral analysis is carried out on the counts of the deep space view (DSV counts).



In this study, we investigate three estimates of noise: We calculate the Allan Deviation on the deep space view counts to obtain the DSV count-noise:

$$\Delta C_{\mathrm{DSV}} = \sqrt{\frac{1}{2(N-1)} \sum_{n=1}^{N-1} \sum_{k=1}^{K} (C_{\mathrm{DSV}_{k,n+1}} - C_{\mathrm{DSV}_{k,n}})^2} \tag{11}$$

At first, the difference in the counts $C_{\mathrm{DSV}}$ from scan line $n$ to scan line $n+1$ is calculated for each view $k$ separately. Then,

the average for all $K=4$ views is taken. Then the total Allan Deviation is computed as the average over all $N-1$ values obtained for the window of $N=300$ scan lines. This estimate of count noise is then translated into a temperature: We deduce the cold NE$\Delta$T by dividing by the gain corresponding to the first of the two adjacent scan lines (equally one could take the gain corresponding to the second one)

$$\mathrm{NE}\Delta\mathrm{T}_{\mathrm{cold}} = \sqrt{\frac{1}{2(N-1)} \sum_{n=1}^{N-1} \sum_{k=1}^{K} \left( \frac{C_{\mathrm{DSV}_{k,n+1}} - C_{\mathrm{DSV}_{k,n}}}{G_n} \right)^2} \tag{12}$$

Similarly, we calculate the warm NE$\Delta$T by replacing the DSV counts by the counts on the on board calibration target (OBCT counts) in eq. 12. These three measures, i.e. the DSV count noise, the cold and the warm NE$\Delta$T, are monitored over the life time of the instruments for each channel. The long time series that are displayed in this study contain data only for every 50th orbit (Fig. 6, 7, 8) in order to avoid a stronger overlapping of symbols and maintain readability.

## 4 Results

### 4.1 Analysis of noise spectrum

The noise spectrum for the different channels has a non-white component that is more or less strong pronounced for the different instruments and years. We present the effects of this mixed spectrum on the calculation of the noise time series. As examples, we pick two orbits from different years of MHS on NOAA-18 (2005: orbit 505, 2007: orbit 4500). The spectrum is calculated for these orbits with the bias function introduced above in eq. 10. The bias functions for each 300-scan line window

are further averaged over the orbit and the four DSV views. In this way, we obtain for each of the two orbits an averaged bias function that is shown for each channel in Fig. 2, together with the simulated bias functions for white and pink noise. For the channels 1,3 and 4, the bias function is close to the one of pure white noise and therefore indicates for these channels a strong white noise component that is dominant over the pink one in the count noise of the DSV. The spectra for channel 2 and 5 look different, though. Both channels show a strong deviation from the pure white noise case, rather indicating a mixture of white

and pink noise for both years.

How far this affects our noise estimates can be deduced from looking at the corresponding periods in time for the actual calculated count noise: In Fig. 3 the time evolution of the DSV count noise for the five channels is shown. In addition to the count noise calculated with the inter-scan line method, we also provide the estimates obtained from the inter-pixel method for the two investigated orbits (red dots). The comparison of both methods' results together with the spectra in Fig. 2, indicates





that both methods agree as long as there is a strong white noise component only (channels 1,3 and 4). Hence, the jump in DSV count noise in late 2007 in channel 3 and 4 is captured by both methods. At this time one can observe sudden jumps in the mean counts as well as a suddenly increased spread of the recorded counts around the mean, not only for the counts in the DSV but also for the OBCT counts. This is probably due to a gain adjustment for channels 3 and 4 in september 2007 (NOAA-OSPO, 2015).

If the noise spectrum is a mixture with a strong pink component, however, as it is the case for channels 2 and 5, the inter-scan line method gives a higher value than the inter-pixel method. This difference in the results of the two methods seems reasonable, since a pink-noise (1/f noise) contaminated signal having larger noise power at smaller frequencies, has variations due to noise on a longer time scale than the inter-pixel time scale. Thinking of the calibration cycle of the instruments, one can imagine the following scenario and consequences for the uncertainty estimation: At the beginning of the earth scan, the signal suffers from a certain unknown portion of noise. Later in the scan, when looking at DSV, the signal from the target (deep space) itself is smaller of course. The portion of noise that contaminates the signal will have changed in the meantime, too. In the case of pure white noise, we know the range of that longer-term change since it will be defined by the standard deviation of the underlying distribution. This standard deviation is described by the value of the inter-pixel count noise. However, in the case of pink noise (or a mixture of white and pink), that longer-term change may have a different magnitude because of the stronger contribution of smaller frequencies to the noise. The inter-pixel noise value cannot capture this larger, longer-term change. Executing all the processing of the measured signal, one obtains the final brightness temperature $T_{a,scene}$. Naturally, this $T_{a,scene}$ is not the real value, but $T_{a,scene}$ is an estimate that will have an uncertainty. If we took the inter-pixel noise value as the uncertainty due to noise, we would underestimate the uncertainty. These longer-term variations between different targets within one calibration cycle are captured in the inter-scan line method (as far as they do not exceed the time scale of two scan lines) and therefore yield a higher value as noise estimate. This possibly significant change in the amount of noise that can happen between the measurements of the earth views and the calibration views due to pink noise should be included in an estimate of uncertainty of the final brightness temperature measurement. Therefore, avoiding underestimation of the uncertainty, we use the inter-scan line method for the calculation of noise.

Exemplarily we investigate the noise spectrum for the different instruments and channels in some chosen orbits and years across their lifetime. Naturally, this investigation cannot fully resolve the evolution of changes in the spectrum, but our analysis provides snapshots of the overall evolution of the spectrum. The AMSU-B and MHS instruments show in their channels either pure white noise or a mixture of white and pink noise. The distribution of this characteristic among the channels is not fixed, however: A certain channel, for example the central water vapour channel 3, does not necessarily exhibit the same noise characteristic in all AMSU-B and MHS instruments. Furthermore, the characteristic may change as well in time. Looking at AMSU-B on NOAA-17 in Fig. 4 for example, channel 3 shows a strong pink component in the year 2006, whereas two years before in 2003 the pink component was less pronounced. This change in spectrum, adding some pink component to the noise, is also captured in our noise estimation by the inter-scan line-method: We detect a higher noise value accounting for the increased level of uncertainty that is due to the increased pink component. This is visible in the corresponding DSV count noise shown in Fig. 5.





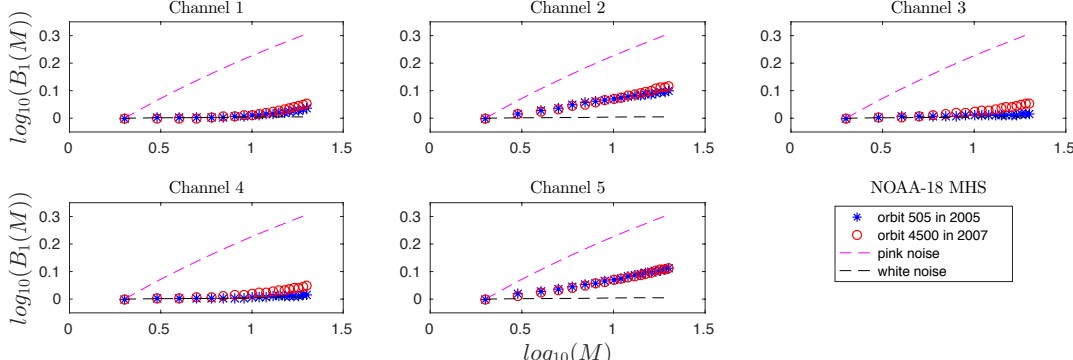

**Figure 2.** The bias functions $B_1(M)$ per channel for the orbits 505 of the year 2005 and 4500 of the year 2007 for MHS on NOAA-18

For the SSMT-2 instruments, the bias function method as we use it here for analysing the noise spectrum does not work properly for all times and channels. The reason for that lies in the absolute count values that are so small for the SSMT-2 that the digitization noise may impact and distort the picture. To improve this bias function method for the usage on data affected by digitization noise, one should simulate the digitization as well as the white and pink noise as has been presented by Mittaz
**?**. Another aspect that impacts even more the noise analysis is the multitude of outliers in the measurements of the SSMT-2 instruments that often disturb the noise estimation. As mentioned above, we applied no filtering in order to get the whole picture of the instruments' behaviour: the instability in the performance of PRT, OBCT and DSV measurements of SSMT2 gets clearly visible in comparison to the other instruments. In the processing of the data to level 1c FIDUCEO FCDRs, those outliers are filtered out and do not contribute to the noise estimation executed on the fly.

## 4.2 Evolution of noise

We provide an overview over the evolution of noise in the different channels over the life time of the instruments (a detailed description of the instruments' performance is given in the appendix). The three measures of noise, i.e. the DSV count noise, the cold and the warm NE$\Delta$T, are displayed for all instruments and channels in Fig. 6, 7 and 8. The DSV count noise (see Fig. 6) is given in absolute counts and is therefore not suited for a comparison of noise levels of different instruments. The
individual instrumental stability of the noise level can be observed very well, however. Looking at channels 3 and 4 of SSMT-2 on F14, one can observe a significant increase of the DSV count noise from 2001 on. A strong degradation of the DSV count noise is visible also for channel 1 of AMSU-B on NOAA-17: from 2007 on, the noise often peaks at almost 10 times higher values that its original one. Channel 3 and 4 of MHS on Metop-A show a rather smooth change over several years: from 2009 to 2012 the DSV count noise smoothly increases. Then it abruptly jumps back to its initial value before increasing smoothly
again. During the years 2014 to 2016 it then decreased again. The DSV count noise of AMSU-B on NOAA-15 and -16 varies only very slightly and smoothly over the life time of the instruments.





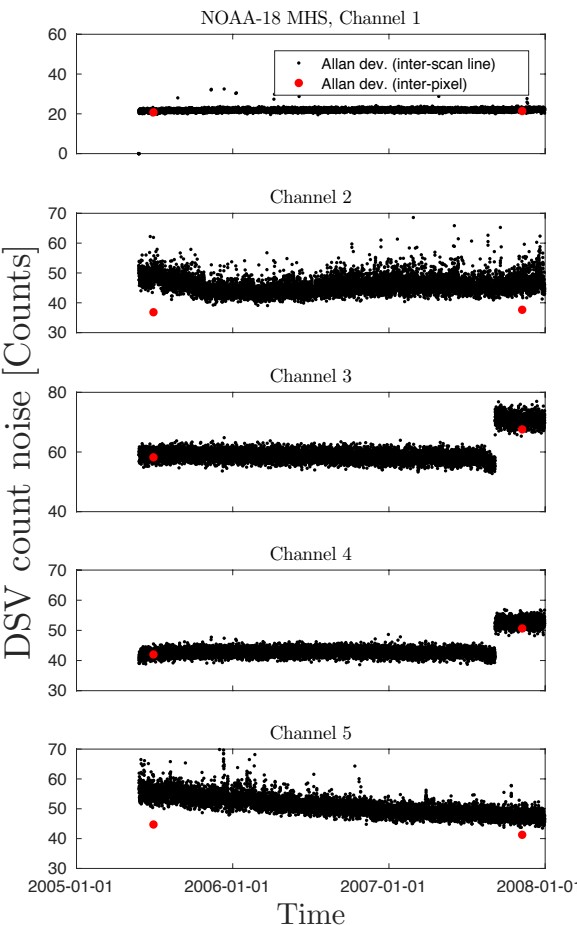

**Figure 3.** The DSV count noise per channel for the years 2005 to 2008 for MHS on NOAA-18. For the orbits 505 of 2005 and 4500 of 2007, the red dots indicate the count noise obtained by the inter-pixel method.





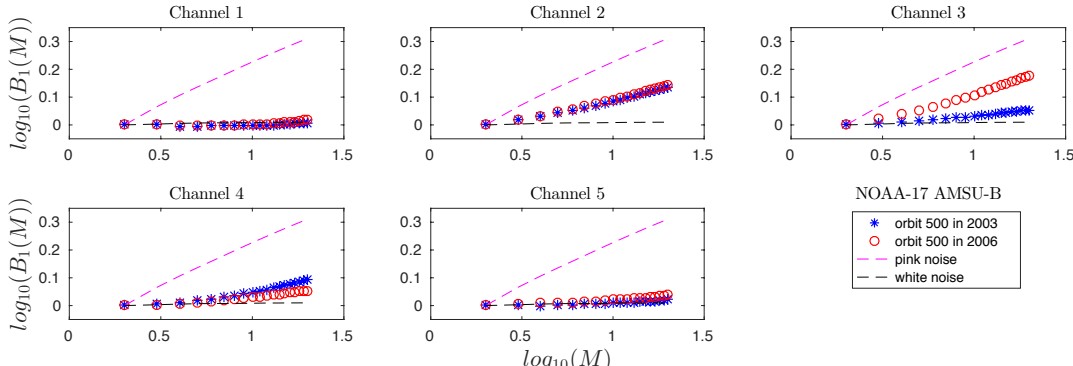

**Figure 4.** The bias functions $B_1(M)$ per channel for the orbits 500 of the year 2003 and the year 2006 for AMSU-B on NOAA-17

Both instruments, however, show a very different picture for the warm and cold NE$\Delta$T. Its evolution is displayed in Fig. 7 and 8. The NE$\Delta$T is influenced by the underlying count noise and the gain used for the conversion to temperature. Therefore, the evolution reflects the interplay of both quantities. The overall increase of NE$\Delta$T therefore relates to an increase of the count noise or a decrease of the gain. The increases in DSV count noise discussed above are quite visible in the cold NE$\Delta$T as well,

e.g. for the channel 1 of AMSU-B on NOAA-17 or for channel 3 and 4 of MHS on Metop-A. The same is valid for the count noise of the internal calibration target view and the warm NE$\Delta$T that is usually about 0.1 K higher than the cold NE$\Delta$T. For the channels 3 to 5 of AMSU-B on NOAA-15 and -16, that showed an almost stable count noise, the cold and warm NE$\Delta$T shows a strong increase over the life time reaching e.g. 5 K in channel 3 in 2010, superimposed with an oscillating pattern. This increase is due to a strong degradation and decrease of the gain that has been observed in (John et al., 2013),too. The

oscillating pattern is also observed in many other measured quantities for these periods and is probably related to the change of the solar-beta-angle as the orbit of the satellite drifts, see appendix and (Zou and Wang, 2011). This changing pattern is also visible for cold NE$\Delta$T of channels 3 and 4 of MHS on NOAA-18 from late 2014 on. However, there is no steady degradation of the gain as for NOAA-15 and -16, such that the cold NE$\Delta$T remains at rather low values. The cold NE$\Delta$T also reflects erratic behaviour of the instrument when the smooth evolution of the quantities is interrupted by sudden jumps: For example,

channels 3 and 4 of MHS on NOAA-19 suffer from an incident in late 2009 where NE$\Delta$T suddenly rises up and lowers again, but stays at an increased level.

## 5   Discussion

In this study we used the Allan Deviation to calculate the evolution of the noise as well as the noise spectrum for the microwave sounders SSMT-2, AMSU-B and MHS in order to assess the quality of the data with respect to uncertainty due to noise.

The analysis of the noise spectrum showed that in some channels there is a significant non-white component that may change during the life time of the instruments. Together with the corresponding periods of count noise evolution in time, the analysis of the spectrum revealed that the inter-scan line method for computing the Allan Deviation is better suited for the purpose of





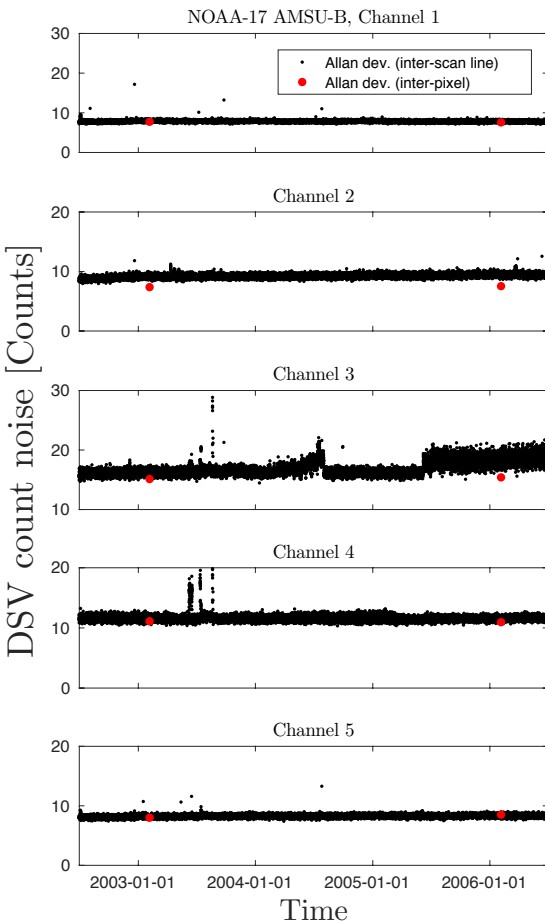

**Figure 5.** The DSV count noise per channel for the years 2003 to 2006 for AMSU-B on NOAA-17. For the orbits 500 of 2003 of 2006, the red dots indicate the count noise obtained by the inter-pixel method.



**Figure 6.** The time evolution of the DSV count noise for the five frequency channels.





**Figure 7.** The time evolution of the cold NEΔT for the five frequency channels.







**Figure 8.** The time evolution of the warm NEΔT for the five frequency channels.



uncertainty estimation than the inter-pixel method that underestimates the uncertainty if a pink noise component is present. Although the analysis of the noise spectrum was carried out on some orbits only, it definitely shows important aspects of the spectrum and its possible evolutions. Nonetheless, a full analysis of the noise spectrum would require a study on all orbits to track the evolution of the spectrum over time.

For the quality assessment of the microwave sounder data, we investigated the evolution of noise (count noise and NEΔT) over the life time of the instruments. The graphical overview we provided with Fig. 6 to 8 on the evolution of the noise gives a first impression of the quality of the data. The various outliers that we did not filter out on purpose indicate problematic periods of the instruments. The actual reasons for the various kinds of outliers are unclear.

Degradation in quality also manifests itself in an increasing cold NEΔT. This degradation can have two reasons: First, the
actual noise level measured in the count noise may have increased. This effect is hardly visible on mission time scales as the count noise is rather stable for most instruments. But on monthly time scales, the effect of increasing and subsequent decreasing of count noise shines through in the changes of cold NEΔT. Yet, the count noise does not cause an overall steady degradation for the investigated instruments. The second possible reason for degradation, however, has a strong impact on NEΔT in the observed cases: If the gain decreases and therefore the measured counts of DSV and OBCT converge, the NEΔT increases
strongly. This reflects that the radiometer sensitivity, which NEΔT is a measure for, strongly degrades and the instrument is not able to distinguish temperatures properly any more. E.g. it can only determine a temperature with an uncertainty of about 5 K, as it is the case for channel 3 of AMSU-B on NOAA-16 in 2010. This effect of gain degradation and increase of NEΔT is visible on both short and long time scales: The pattern induced by the change of the solar-beta-angle modifies NEΔT on monthly time scales and an overall continuous degradation of the gain causes a steady increase of cold and warm NEΔT as
seen for NOAA-15 and NOAA-16.

As intuitively obvious, an ageing satellite or sensor may degrade since its components have a limited life time. Accordingly, one can observe this degradation for many of the considered instruments. An interesting fact here is the different evolution for the different channels: when the three water vapour sounding channels severely degrade, the lower peaking channels may be unchanged, i.e. they may show no sign of ageing. Or, there are events that are visible in all channels, but only show long
lasting impact on certain channels. For the newer satellites, some adjustments were made during operation and thus prevented the instruments from degradation and kept them on an acceptable noise level. The lowest and most stable noise, but also the shortest data record so far, has the MHS instrument on board the METOP-B satellite.

As an easy-to-use tool for information on noise we provided plots of the time evolution for all individual instruments of this microwave sounder family. These plots may help to decide on the usability of the data for a certain application. They were
given for the DSV count noise, the warm and the cold NEΔT. The user of the data has to decide which level of uncertainty his or her product generation might still bear and which threshold of NEΔT he or she would set to limit the uncertainty. As further result, we provide a chart in Fig. 9, which shows the periods of data for a threshold of cold NEΔT < 1 K.

For atmospheric product retrieval, Fig. 7 and 8 together with eq. 5 can be used to estimate the correct scene NEΔT. Since warm and cold NEΔT typically differ by only approximately 0.1 K, a reasonable approximation would be also to simply use
the warm NEΔT as estimate for the scene NEΔT.



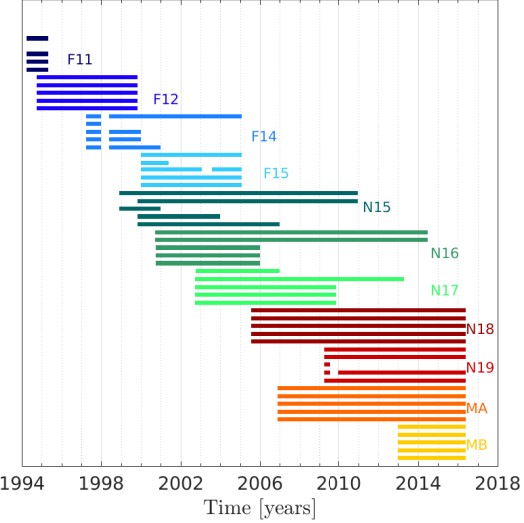

**Figure 9.** Usable microwave data records with cold NEΔT <1K. The five bars per satellite correspond to the channels 1 to 5 (from top to bottom).

## 6    Conclusions

The results of our study provide the user with information on uncertainty due to noise that he or she has to expect when using the data sets of the microwave sounders SSMT-2, AMSU-B and MHS.

The chart in Fig. 9 reveals the possibility to concatenate the available data for constructing gap-less long time series since
5   1994 at a noise level below 1 K for all frequency intervals that the instruments cover. This is of major interest for climate researchers who need long time series with low noise levels in order to investigate possible trends.

Apart from the stand alone results as information content for users of these microwave sounders' data, our analysis is of direct use for the FIDUCEO project: The method for estimating the count noise for the DSV and OBCT will be used in the processing of level 1b to level 1c FIDUCEO FCDR in order to provide on the fly input values for the uncertainty propagation.
10   This FCDR will provide a field-of-view-wise estimate of uncertainty in brightness temperature due to count noise for every scan line and orbit. Besides, the FCDR will contain extensive information that will further close the gap of lacking information on uncertainty.

*Data availability.* The data from SSMT-2, AMSU-B and MHS are available from NOAA CLASS, http://www.class.ngdc.noaa.gov/saa/products/catSearch



## Appendix A:  Sensor time series

In the following we investigate the stability of the individual instruments flying on different satellites by looking at the long time series of the above mentioned observables, mostly at the cold NE$\Delta$T as indicator of the overall noise. For every channel, we display the cold NE$\Delta$T over all considered missions in Fig. 7 from 0.1 to 5 K. We state which data we would definitely suggest to exclude, based on the rather high threshold of 1 K. The remaining useful periods are displayed above in Fig. 9. We are interested in long term evolutions in the sensor or sudden incidents impacting the instrument. Hence, the normal orbital variations are not investigated further, since their effect on the cold NE$\Delta$T, even in case of stronger changes, is only very small by construction.

### A1    DMSP-F11 (SSMT-2)

The DMSP-F11 was launched in 1991. The NOAA CLASS data set starts at 01 April 1994 and ends on 24 April 1995 with some data gaps of several days or weeks. The time record exhibits some issue: Sometimes the time stamp indicating the seconds of the day is zero (without a change of day) or has values larger than 86400 s. The corresponding scan lines are excluded in our processing and do not enter the time series.

#### A1.1    OBCT temperature (2 PRT)

Both PRT sensors show normal behaviour throughout the time range. The temperature on the black body changes in an interval of about 5K around 290 K.

#### A1.2    Channel 1, 2

Channel 1 has a stable gain and a low cold NE$\Delta$T of 0.2 K over the whole time range (see Fig. 7a, black line). Channel 2, however, is damaged from start on: the gain is constantly zero as the signal for the OBCT and DSV counts is the same. Hence, it is of no use for research. The cold NE$\Delta$T has infinite values and therefore does not appear in Fig. 7b.

#### A1.3    Channel 3, 4, 5

The gain is stable at about 10 Counts K$^{-1}$, except for some erroneous outliers between -5 and 10 Counts K$^{-1}$ (very similar values for all three channels). In November 1994, there is a complete orbit of bad outlier data spreading between -5 and 10 Counts K$^{-1}$. The cold NE$\Delta$T is quite stable at around 0.3, 0.4 and 0.5 for channels 3, 4 and 5 respectively - except for the corresponding outliers of the gain (Fig. 7c-e, black line). From late 1994 and early 1995 on, cold NE$\Delta$T of channel 3 and 5 shows more frequently higher values around 1.3 K. This is due to a most peculiar aspect: There are jumps up and down in the OBCT and DSV counts within an orbit from the year 2001 on (it already appears before, but rather seldom): The orbital change has the expected smooth shape before it suddenly jumps to a higher or lower level and there continues its course. The origin of those jumps is unclear.





## A2 DMSP-F12 (SSMT-2)

The second SSMT-2 instrument was brought to its orbit on 28 April 1994. The NOAA CLASS data set reaches from 13 October 1994 to 08 January 2001 with some data gaps of several weeks.

Beside the time record issues mentioned for F11, the instrument on F12 revealed more wrong time stamps for many data points: The time stamp goes back to some hours before the actual time and therefore produces artificial abrupt rises and drops in the time evolution of the observables. Hence, additionally to the filter used for F11, we use a second one excluding all data whose time stamp is smaller than the previous one.

### A2.1 OBCT temperature (2 PRT)

The PRT sensors do not show any peculiarity, except for several groups of outliers in 1994 (around 288 K) and more widely distributed outliers in 1999. Both PRTs show slight oscillatory changes in the black body temperature of about 4 K around an increasing mean of 300 to 304 K. In 1994 and later in 1999 there are several groups of outliers.

### A2.2 Channel 1, 2

The lower peaking channels show the same behaviour as the water vapour channels described below, with similar values. They cannot be used for research purposes after 1999 either (see Fig. 7a,b, violet line).

### A2.3 Channel 3, 4, 5

Apart from outliers, the gain is stable until 1999 at around 10 or 9 Counts K$^{-1}$ for channels 4,5 and 3 respectively. The same holds for the DSV count noise and the cold NE$\Delta$T (0.43, 0.34, 0.38 K for channel 3,4,5). From later 1999 on, there are very many outliers that are rather widely spread such that cold NE$\Delta$T also reaches above 5 K for those data points and the remaining line of cold NE$\Delta$T around 0.4 K appears quite thin (see Fig. 7c-e, violet line). This makes the water vapour channels less suited for research purposes.

## A3 DMSP-F14 (SSMT-2)

The third SSMT-2 instrument was only launched on board the DMSP-F14 on 10 April 1997. The NOAA CLASS data set starts on 28 April 1997 and ends on 18 January 2005

### A3.1 OBCT temperature (2 PRT)

The black body temperature slightly oscillates with a period of about 6 months around a decreasing mean from 294 K to 292 K in 2005. Both PRT sensors agree within the random uncertainties throughout the investigated time frame.





### A3.2 Channel 1, 2

The low-peaking channels show a similar behaviour as the water-vapour channels until 1998, when the instrument suffers from several issues, described in detail for channels 3 to 5. Channel 1 recovers from the critical 1998-phase and has a very low cold NE$\Delta$T at the level of the pre-1998 values of 0.3 K (Fig. 7a, blue line). Channel 2, however, does not recover after May 1998:

Instead, the signal of the OBCT and DSV approach each other resulting in a strongly increasing cold NE$\Delta$T surpassing 1.5 K already at the end of 1998. Afterwards, it even reaches 8 K before decreasing slightly again, but ever staying above 6 K (Fig. 7b, blue line). Channel 2 is therefore only usable for 1997. Both channels also show the jumps of unclear origin, already mentioned for F11, but to a lesser extent than the water vapour channels described below.

### A3.3 Channel 3, 4, 5

The gain remains stable at 10 and 9 Counts for channel 4 and 3, 5, respectively, throughout the lifetime. However, during the first half of 1998 the instrument suffers from some incidents: several additional levels of gain emerge since the OBCT and DSV counts show extensive jumps. Thus it appears as if the levels of OBCT and DSV counts signal split and approach each other. The resulting gain levels are lower than the original stable value or even close to zero which leads to many high peaks of cold NE$\Delta$T of even >1000 K (not visible in Fig. 7c-e). Data from this period, i.e. January until May 1998, should not be

used. Apart from this period the cold NE$\Delta$T is quite stable with slight changes around 0.5 K for channels 4 and 3, or 0.4 K for channel 5. In 2001, however, cold NE$\Delta$T increases above 1.5 K for channel 4 (4.5 K for channel 3 and 0.7 K for channel 5) and stays at this high level (see Fig. 7d, blue line). This corresponds to the development of the DSV count noise: after 1998, the DSV count noise also increases from initially 4 Counts to 15 Counts, slightly at first, then stronger in 2001 (even 30 Counts in channel 3, 8 Counts for channel 5). Later on, the values change around this increased level. To some extent, this correlates

with the more frequent appearance of the jumps in the OBCT and DSV counts within an orbit as mentioned already for F11. Due to the described increase of the cold NE$\Delta$T, channel 3 and 4 should not be used from year 2001 on. Channel 5 might be used with caution due to higher uncertainty resulting from the jumps.

### A4 DMSP-F15 (SSMT-2)

On 12.12.1999 the DMSP-F15 satellite was launched carrying the last SSMT-2 instrument. The NOAA CLASS data set en-

compasses the measurements from 24 January 2000 to 18 January 2005.

### A4.1 OBCT temperature (2 PRT)

Throughout the considered time frame, both PRT sensors indicate a stable, only slightly oscillating black body temperature around an increasing mean of 295 to 298 K.





### A4.2   Channel 1, 2

After a stable phase at the beginning of the mission, the gain gets slightly unstable for channel 1 and smoothly increases from 7 Counts $K^{-1}$ to 9 Counts $K^{-1}$ before decreasing to 5 Counts $K^{-1}$. Accordingly, cold NE$\Delta$T increases from 0.6 K to 0.8 K. In February, March and September 2003, channel 1 suffers from very large noise >5 K. These periods should be

excluded. Furthermore, in 2003, there is a second level of cold NE$\Delta$T values which the measurement jumps to and off and which increases from 3 to 4 K (Fig. 7a, light blue line). This pattern can be seen in the DSV count noise as well and relates to the same jumps of unclear origin as those mentioned below for the channel 3 to 5. These are also visible in the OBCT and DSV counts of channel 2. However, the gain for channel 2 already decreases from 2001 on when the OBCT and DSV signal become similar. Accordingly, cold NE$\Delta$T rises and even reaches 5 K. It does not decrease below 2.3 K afterwards (Fig. 7b

light blue line). Hence, channel 1 could be used with caution due to some higher uncertainty, whereas channel 2 is of no use due to its large noise.

### A4.3   Channel 3, 4, 5

The gain is quite stable at a constant value of 7 Counts $K^{-1}$ for channel 4 (8 Counts $K^{-1}$ for channels 3 and 5), but has many outliers even down to a negative gain of -3 Counts $K^{-1}$. cold NE$\Delta$T is mostly stable at 0.5 K (0.6 K for channels 3 and 5).

In 2003, cold NE$\Delta$T temporarily increases in channel 3 to 1.5 K, but decreases again to 0.8 K (see Fig. 7c, light blue line). Channels 4 and 5 remain quite stable (Fig. 7d-e, light blue line). However, from start on, the jumps of unclear origin, mentioned for the surface channels above and for F11 and F14, appear in channel 4 and 5 and make the DSV count noise as well as the cold NE$\Delta$T change suddenly between two courses.

### A5   NOAA-15 (AMSU-B)

On 13 May 1998 the NOAA-15 satellite was launched having the first AMSU-B sensor on board as subunit of the AMSU instrument. The operational data start on 15 December 1998. The instrument was turned off on 28 March 2011 (NOAA-OSPO, 2015), but already in late 2010 the data are too noisy to be used. Here, we investigate the NOAA CLASS data set from start of operational data until the end of 2010. AMSU-B was turned off due to problems with the scan motor making measurements impossible. However, there are still data records being sent to Earth which cannot be used, of course, since these contain no

measurement data but random numbers.

The NOAA-15 satellite started with an LECT of about 19:30, reached about 16:30 in 2010 and drifted back to 18:00. Its quick orbital drift over its lifetime impacted on the AMSU-B instrument: A characteristic pattern of peaks and drops becomes visible in the time evolution of many observables from 2002 on (see also NE$\Delta$T in Fig. 7 and 8). The same pattern can also be seen for the Microwave Sounding Unit (MSU) instrument on the earlier NOAA-14 satellite (Grody et al., 2004), for the

AMSU-A on NOAA-15 (Zou and Wang, 2011) as well as on the AMSU-A and -B on board the successor satellite NOAA-16 (see below) which has already experienced the same strong orbital drift as other NOAA satellites. In (Zou and Wang, 2011), yet focussing on AMSU-A, a connection of this pattern to a changing solar beta angle due to orbital drift is seen. This angle





is defined as the angle between the vector from Earth to Sun and the orbital plane of the satellite. Hence, a changing angle will influence the exposure of the instrument to the sun and may therefore impact its performance. An investigation of this is beyond the scope of this overview of microwave data.

The AMSU-B on NOAA-15 also suffered from the radio frequency interference (RFI) with channels 2 and 4 being impacted
most. It introduced a scan dependent bias that also effected the deep space view beside the 90 Earth views. The impact was not constant in time, however. For example, in the period of October 1998 to September 1999, the measurements are biased for half the orbit before returning to normal behaviour for the rest of the orbit (Atkinson, 2001). This is also visible in the cold NE$\Delta$T of channels 2, 4 and 5 (see Fig. 7b,d,e, dark green line).

### A5.1   OBCT temperature (7 PRT)

From start on, the black body shows strong variations of temperature (5 to 8 K) on monthly scale. Moreover, there are many drops to 262 K which are probably related to the PRT sensors. All seven sensors mostly agree throughout the life time, apart from some events where they drop or jump to different temperature levels. There are also many randomly distributed outlier values of the different PRT sensors. From 2002 on, the orbital drift induced the changing pattern mentioned above becomes clearly visible and remains till the end of the data set.

### A5.2   Channel 1, 2

The counts for the OBCT and DSV are quite stable, except for small changes on monthly scale. However, the counts often drop to zero (either for both targets or for one of them) which results in constant levels of outliers in the gain at -60, 0 or 100 Counts K$^{-1}$. Yet, apart from other random outliers the gain is mostly stable at its initial value of 30 and 20 Counts K$^{-1}$ for channel 1 and 2 respectively. The changing pattern mentioned above becomes more pronounced in the course of time, but as
the OBCT and DSV counts almost change accordingly there are only very small changes in the gain ($\sim 1$Count K$^{-1}$) and no decline. The cold NE$\Delta$T remains quite stable at 0.25 K (channel 1) and 0.6 K (channel 2), see the dark green line in Fig. 7a and b, respectively. Filtering out the scan lines of outlier values, and excluding channel 2 from start until November 2000, when a phase of unstable cold NE$\Delta$T ends, will provide a useful data set.

### A5.3   Channel 3, 4, 5

The water vapour channels are subject to more quality issues: From start on, one can observe slowly decreasing counts for the DSV signal and quicker decreasing for the OBCT counts. For the first years until end of 2001, the resulting gain has still acceptable values and cold NE$\Delta$T is about 1 K for channel 3 or 0.8 K and 0.6 K for channels 4 and 5, respectively. From 2002 on, however, the changing pattern as seen in the black body temperature shines through also to cold NE$\Delta$T and the degradation gets stronger: The recorded signal for OBCT and DSV approach each other until the gain becomes very small (below 6 Counts
K$^{-1}$ for an initial value of 20 Counts K$^{-1}$) and, consequently, the cold NE$\Delta$T rises above 2.5 K. Finally in the middle of September 2010, the gain drops to zero resulting in NAN values for cold NE$\Delta$T. Data should not be used for channel 3 from





2001 on, for channel 4 from 2004 on and for channel 5 from 2007 on as the cold NEΔT increases beyond 1 K (see Fig. 7c-e, dark green line).

## A6 NOAA-16 (AMSU-B)

The second AMSU-B instrument was sent to space on board the NOAA-16 satellite on 21 September 2000. The operational
data started on 20.03.2001. Finally, NOAA-16 was decommissioned on 09 June 2014. Compared to its predecessor, NOAA-16 was exposed to an even stronger orbital drift from about 14:00 to 22:00 LECT (see Fig. 1). In 2007, the earlier mentioned changing pattern for the observables emerges, probably related to the solar-beta-angle (see above, (Zou and Wang, 2011)). It is visible in NEΔT, too (see Fig. 7 and 8).

### A6.1 OBCT temperature (7 PRT)

The black body temperature only shows small oscillations on monthly scale which reach about 4 K in late 2006, though. As
for NOAA-15, the PRT sensors also often drop to 262 K. There are also periods of months, where the PRT sensors differ about 10 K for several orbits. Then, from October 2007 on, the variations in the overall evolution become more severe as the strong changing pattern becomes visible with an amplitude of 5 to 10 K. In 2012, the pattern ceases and only small changes around 288 K can be seen.

### A6.2 Channel 1, 2

The low peaking channels show quite acceptable data having a cold NEΔT of 0.3 K. Nonetheless, over the whole lifetime, the OBCT and DSV counts often drop to zero or jump to other quite stationary levels (especially from 2004 on). This is transported to the gain and also causes outliers of up to 2 K in cold NEΔT. In Channels 1 and 2 the changing pattern is very faint and only changes the gain about ±1%. Therefore, the cold NEΔT also appears stable on the scale of Fig. 7a,b, green line).

### A6.3 Channel 3, 4, 5

At first, the gain is rather stable for the three water vapour channels. A slight decreasing starts in early 2001 after higher orbit-to-orbit variations that can be seen in OBCT and DSV counts, as well. In 2002, the OBCT counts start to decrease quicker than the DSV counts, and hence the gain decreases continuously. Four years later, in 2006, the gain has decreased from initially 22 Counts K$^{-1}$ down to 9 Counts K$^{-1}$ in channel 3 (the other channels show a similar evolution) and cold NEΔT has risen from
0.6 K to 1.4 K. The degradation for the three channels continues further as the the gain decreases (OBCT and DSV counts getting close to one another) and cold NEΔT increases. From late 2007 on, the changing pattern shines through in the counts and the cold NEΔT (see Fig. 7c-e, green line) reaches 18 K in 2011, when the gain approaches zero, and increases beyond 50 K in 2014 as the signal recorded for the OBCT and DSV is basically the same. Doing a two point calibration is not sensible at this stage and produces completely useless data due to absurdly high noise with cold NEΔT > 10 K. One should stop using
NOAA-16 data with the end of 2005 when cold NEΔT surpasses 1 K and degradation keeps advancing in channels 3-5.



## A7    NOAA-17 (AMSU-B)

On board NOAA-17 the last AMSU-B instrument was launched on 24 June 2002. Its operational data set starts on 15.10.2002 and ends on 10 April 2013. NOAA-17 drifted from about 22:00 to 19:00 LECT over its mission (Fig. 1).

### A7.1    OBCT temperature (7 PRT)

5    The seven PRT sensors indicate a stable black body temperature softly oscillating on yearly scale around 285 K (slightly increasing to 287 K). As for the other AMSU-B instruments, the PRT measurements also often drop to 262 K. In 2010, the overall evolution remains, but the measured values of the seven sensors jump between discrete levels and follow the overall evolution with different constant offsets. There also appear strong peaks from 2011 on, a sharp drop to 275 K in early 2013 and increase again.

10    ### A7.2    Channel 1, 2

Till 2007, channel 1 has a stable gain, cold NE$\Delta$T and DSV count noise. Then, sharp peaks (of factor $> 4$ to stable noise value) appear in the DSV count noise. Later the peaks reach even a value of factor 10 times the stable noise value and outliers even factor $> 50$. Moreover, the peaks become more frequent such that the underlying constant DSV count noise of initially 8 Counts becomes less visible. Hence, channel 1 punctually gets very noisy (cold NE$\Delta$T peaks reach up to 5 K) due to the DSV 15    count noise that transfers to the overall cold NE$\Delta$T, see Fig. 7a, light green line. The gain is also impacted from the high DSV count noise peaks, since the DSV counts apparently have a larger variation that becomes visible in jumps and drops of the gain to certain levels whilst keeping the overall initial value of 24 Counts $K^{-1}$. Channel 2 shows a similar behaviour, though less pronounced, i.e. the frequency of the appearing peaks is smaller (see Fig. 7b light green line). Filtering out the scan lines of outlier values will lead to a usable data set for channel 2. Channel 1 also needs filtering, but from 2007 on, one should not use 20    the data at all, since it gets too noisy as described in the beginning of the paragraph.

### A7.3    Channel 3, 4, 5

Apart from small jumps and drops in channel 3 and 4 in 2003 and 2004, all three channels have stable cold NE$\Delta$T values of 0.85 K, 0.7 K and 0.8 K, respectively. In December 2009 however, a sharp drop of both OBCT and DSV counts results in a gain of almost zero and a huge cold NE$\Delta$T of 2000 K or infinite (NAN) values (see Fig. 7c-e, light green line). From December 25    2009 on, the NOAA-17 AMSU-B data for the sounding channels cannot be used for any research questions.

## A8    NOAA-18 (MHS)

The first MHS instrument has been installed on board the NOAA-18 satellite launched on 20 May 2005. The operational data set starts on 30 August 2005. The mission is still ongoing, however, our data set for investigation ends in May 2016. From its start until May 2016 it drifted from 14:00 LECT to 18:00.





### A8.1  OBCT temperature (5 PRT)

The five PRT sensors agree on the slight oscillations on yearly scale of the black body temperature around 284 to 287 K. Apart from a few outlier values of several PRTs, the measurements are quite stable and show a stable black body temperature. How-ever, in August 2014, the strong changing pattern as seen for the NOAA-15 and 16 satellites emerges and leads to maximum
(minimum) temperature of 298 K (270 K). This pattern is still visible at the end of the used data set in May 2016.

### A8.2  Channel 1, 2

Apart from outlying values, both channels 1 and 2 have a stable gain and cold NEΔT around 0.14 K and 0.36 K, respectively, over the life time (see Fig. 7a,b, dark red line). The changing pattern visible in the black body temperature is only prominent in the OBCT and DSV counts that change accordingly, thus resulting in a stable gain.

### A8.3  Channel 3, 4, 5

A first, all three channels show a stable gain (in the range of 140 Counts $K^{-1}$), with small discrete jumps and drops. The orbital variation around the mean is larger than for channels 1 and 2, often about $\pm 10$ Counts $K^{-1}$, and also shows changes over the years. Channel 5 has very large orbital variation in 2011 and 2012 and also significant changes in the DSV count noise for these periods, but then it is suddenly reduced by a factor 20 by controlled gain adjustment (Bonsignori, 2007). Thus, channel 5
gets less variable from 2013 on. The changing pattern gets apparent in the gain in 2014: its strongest impact is on channel 3 (up to 90 Counts $K^{-1}$ within a month), then 4 and at last 5 where it is hardly visible. Cold NEΔT is also stable at first (0.5 K, 0.4 K, 0.3 K, for channels 3 to 5), but also shows the jumps as the gain and increases slightly until, in 2014, the changing pattern becomes visible and increases or decreases cold NEΔT (Fig. 7c-e, dark red line). Temporarily, cold NEΔT reaches 0.95 K in channel 3 (0.8 K for channel 4, whereas channel 5 is still stable since 2013 at 0.3 K). It is a usable data set, but one should be
aware of the temporarily increased noise and therefore larger uncertainty for all three channels. Channel 5 has least problems from 2013 on.

### A9  NOAA-19 (MHS)

On 06 February 2009 the NOAA-19 satellite was launched carrying the second MHS instrument. The operational data starts on 02 June 2009. So far, NOAA-19 drifted from 14:00 to 15:00 LECT. It is still operational.

### A9.1  OBCT temperature (5 PRT)

All five PRT sensors measure the same stable temperature of the black body, oscillating slightly on yearly scale around 285 K.

### A9.2  Channel 1, 2

Throughout the life time both channels are stable and have a constant cold NEΔT of 0.13 K and 0.33 K respectively. In Fig. 7a,b, the corresponding red line is directly behind the orange one of METOP-A.





### A9.3  Channel 3, 4, 5

Channel 3 and 4 begin stable, but show erratic behaviour in July 2009: The OBCT and DSV signal suffer from major incidents, resulting in a strongly diminished gain. Following the drop in the gain, cold NE$\Delta$T increases from 0.5 K to 3.4 K in channel 3 (Fig. 7c, red line). Yet, channel 4 recovers from the incidents in 2009 and then remains stable at 0.58 K (Fig. 7d, red line). Channel 5 is stable throughout the mission having a low cold NE$\Delta$T of 0.27 K (Fig. 7e, red line). From the data set of NOAA-19, channel 3 should not be used.

### A10  METOP-A (MHS)

The third MHS instrument was carried to orbit on board the METOP-A satellite launched on 19 October 2006. The operational data start on 15 May 2007. The instrument is still active. Unlike the NOAA satellites, the METOP satellites do not exhibit orbital drift. Their local equator crossing time remains stable at 21:30.

### A10.1  OBCT temperature (5 PRT)

The temperature of the black body is quite stable over the mission so far and shows small variations on a three-monthly scale around 283 K. There are a few orbits with outlier values, rather in the first years of the mission and there is a larger data gap in spring 2014.

### A10.2  Channel 1, 2

Both channels do not show any anomalies and remain stable at their initial cold NE$\Delta$T values of 0.13 K and 0.31 K respectively (see Fig. 7a,b, orange line). The latter one increases slightly to 0.34 K in 2016.

### A10.3  Channel 3, 4, 5

The gain is constantly adjusted during operation to correct for decreasing and increasing and keep it in certain limits (Bonsignori, 2007). Overall, the resulting cold NE$\Delta$T is quite stable around 0.5 K or 0.6 K for channel 3, peaking at 0.7 K in late 2011. For channel 4 there is a slightly lower noise of 0.3 K, peaking at 0.5 K in late 2011. Channel 5 is stable throughout the mission with low cold NE$\Delta$T of 0.27 K (see Fig. 7c-e, orange line). As for channel 5 of the MHS instrument on NOAA-18, the DSV count noise changes over the mission in channel 3 and 4. This is visible in the cold NE$\Delta$T as well.

### A11  METOP-B (MHS)

On 17 September 2012 the METOP-B satellite was launched having the fourth MHS instrument on board. The first operational data are available for 29 January 2013 when it replaces the METOP-A for operational purposes (WMO, 2016). The mission is envisaged to end after 2018. As METOP-A, METOP-B has no orbital drift either.



### A11.1 OBCT temperature (5 PRT)

Until the end of the considered time frame (May 2016), the temperature of the black body varies with an amplitude of about 2 K on a three-monthly scale around 281 K. There are only four events of outlier values so far.

### A11.2 Channel 1, 2

A small decrease of the gain can be observed for channel 2. However, this degradation is always corrected for by adjusting the gain and resetting it to higher values. The cold NEΔT is stable at 0.18 K for channel 1 and 0.36 K for channel 2 (see Fig. 7a,b, yellow line).

### A11.3 Channel 3, 4, 5

The adjustment of the gain to keep it at a quasi constant level is also prominent for channels 3 to 5 (with the smallest adjustments for channel 5). The cold NEΔT is stable at 0.35 K, 0.27 K and 0.25 K for channels 3, 4 and 5 respectively (see Fig. 7c-e, yellow line).

*Author contributions.* I. Hans developed the MATLAB code for the processing and noise determination and analysis, carried out the analysis and prepared the manuscript. M. Burgdorf supported the development of the MATLAB code for reading the raw data, contributed to the discussion and manuscript. V. John created figure 1, contributed to the discussion and manuscript. J. Mittaz suggested the method for noise analysis, advised the analysis, contributed to the discussion and manuscript. S. Buehler accompanied and advised the investigation, contributed to the discussion and manuscript.

*Competing interests.* The authors declare that they have no conflict of interest.

*Acknowledgements.* I.Hans, M. Burgdorf, V. O. John, J. Mittaz and S. A. Buehler greatfully acknowledge support from the FIDUCEO project ("Fidelity and Uncertainty in Climate data records from Earth Observation") which has received funding from the European Union's Horizon 2020 Programme for Research and Innovation, under Grant Agreement no. 638822. V. O. John was also supported by the U.K. Department of Energy and Climate Change (DECC) and Department of Environment, Food and Rural Affairs (DEFRA) Integrated Climate Programme (GA01101) and the EUMETSAT CMSAF. S. A. Buehler was also supported through the Cluster of Excellence "CliSAP" (EXC177), Universität Hamburg, funded through the German Science Foundation (DFG). The authors would like to thank Oliver Lemke for helpful tips on reading the raw data records.





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
