# Peer review of "Noise performance of microwave humidity sounders over their life time"

_Atmospheric Measurement Techniques, 2017_

## Referee Comment (RC1) · Anonymous Referee #1 · 20 Sep 2017

Summary:

This study provides an assessment of uncertainties in the data record of the microwave humidity sounders due to random noise. The study is motivated by the fact that a time varying noise estimate for each sensor is not readily available to data users. They use the allan variance method to calculate the noise in both the cold and warm NEDT. The result is a useful time series of NEDT for all of the legacy sensors, which can inform long-term data records.

The methods are very clearly laid out. No notable flaws are evident. I have only a few minor comments on the paper.

Specific comments:

[Figure]

Line 5: NEdT is a measure of precision (or noise). Accuracy is bias, in this case due to calibration. Precision and accuracy are different and independent quantities. Here you evaluate the noise so you need to change 'accuracy' to 'precision'.

It would be nice for an unfamiliar reader such as myself to have a table that maps channel number into frequency, bandwidth, prelaunch NEDT.

Page 11, line 5: missing date fro reference ('?')

Figures 3/5: Why do you just show the two red pixels per panel? Is it for clarity? Please comment in caption.

———————————————

---

## Referee Comment (RC2) · Anonymous Referee #2 · 13 Oct 2017

General Comments

This paper is well-written and presents the results of a noise analysis of microwave sounders. The data from these instruments are widely used and this paper provides a valuable resource for users to determine whether data from these instruments should be trusted. The authors do a thorough job explaining the background and the results, and I only have a couple minor comments.

Specific Comments

Page 6, first paragraph (and in some other places) I understand what you mean by saying "four views on the OBCT... four views on the deep space..." But I find it a bit confusing to call those "views" since you also refer to the "deep space view" (singular).

Same thing with saying "90 earth views". Perhaps call it something else like "samples".

"White noise" and "pink noise" are mentioned but I couldn't find a definition for what they are. It may be good to include a brief definition for people not familiar with those terms.

---

## Author Response (AR1)

**Authors' response to Anonymous Referee 1 of "Noise performance of microwave humidity sounders over their life time" by Imke Hans et al.**

*Referee 1:Summary: This study provides an assessment of uncertainties in the data record of the microwave humidity sounders due to random noise. The study is motivated by the fact that a time varying noise estimate for each sensor is not readily avail-*
5 *able to data users. They use the allan variance method to calculate the noise in both the cold and warm NEDT. The result is a useful time series of NEDT for all of the legacy sensors, which can inform long-term data records. The methods are very clearly laid out. No notable flaws are evident. I have only a few minor comments on the paper.*

Thank you very much for you comments on our manuscript. I address them in the following:

*Referee 1: Specific comments: Page 1, line 5: NEdT is a measure of precision (or noise). Accuracy is bias, in this case due to calibration. Precision and accuracy are different and inde- pendent quantities. Here you evaluate the noise so you need to change "accuracy" to "precision".*
You are correct. I will change "accuracy" to "precision".

*Referee 1: It would be nice for an unfamiliar reader such as myself to have a table that maps channel number into frequency, bandwidth, prelaunch NEDT.*
I agree on that. It is nice to have such a table directly at hand in the paper. I will include this in the manuscript.

20 *Referee 1: Page 11, line 5: missing date fro reference (?)*
Corrected.

*Referee 1: Figures 3/5: Why do you just show the two red pixels per panel? Is it for clarity? Please comment in caption.*
The red dots indicate the noise in DSV counts calculated with the inter-pixel method. We applied this method exemplarily only
25 for these two orbits for which we investigated the spectrum as well (orbits 505 of 2005 and 4500 of 2007 for Figure 2 and 3 and orbits 500 of 2003 of 2006 for Figure 4 and 5). We show the two red dots as a comparison to the results from the inter-scanline method (black dots). I will adapt the captions to make this clearer.

**Authors' response to Anonymous Referee 2 of "Noise performance of microwave humidity sounders over their life time" by Imke Hans et al.**

*Referee 2: General Comments This paper is well-written and presents the results of a noise analysis of microwave sounders. The data from these instruments are widely used and this paper provides a valuable resource for users to determine whether*
5  *data from these instruments should be trusted. The authors do a thorough job explaining the background and the results, and I only have a couple minor comments.*

We would like to thank you very much for your comments on our manuscript. We address them in the following:

10  *Referee 2: Specific Comments: Page 6, first paragraph (and in some other places) I understand what you mean by saying "four views on the OBCT ... four views on the deep space..." But I find it a bit confusing to call those "views" since you also refer to the "deep space view" (singular). Same thing with saying "90 earth views". Perhaps call it something else like "samples".*

You are right, this might be confusing. However, these four measurements of deep space or OBCT (or the 90 for Earth) are
15  very often referred to as "views" in literature. I would therefore stick to this term. But I will be more consistent to talk about the four deep space views in plural, and use the singular if speaking of one particular view only.

*Referee 2: "White noise" and "pink noise" are mentioned but I couldn't find a definition for what they are. It may be good to include a brief definition for people not familiar with those terms.*
20  I will include the short definitions: white noise (constant power spectral density), 
[revised manuscript text omitted]